# Optimization and Simulation of Extrusion Parameters in Polymer Compounding: A Comparative Study Using BBD and 3LFFD

**DOI:** 10.3390/polym17131719

**Published:** 2025-06-20

**Authors:** Jamal Alsadi

**Affiliations:** Faculty of Engineering, Jadara University, Irbid 21110, Jordan; jamal.alsadi@ontariotechu.net or j.alsadi@jadara.edu.jo

**Keywords:** polycarbonate, interaction for processing parameters, optimization, simulation, 3LFFD and BBD, regression models, microscopic characterization

## Abstract

Many research studies have looked at process characteristics to improve color choices and create more simulation-accurate models. This research evaluated the processing factors speed (Sp), temperature (T), and feed rate (FRate) and employed two response surface techniques, the three-level full-factorial design (3LFFD) and Box–Behnken design (BBD), to optimize uniform processing settings. An experimental approach was employed to optimize process parameters while holding all other variables constant. The Design Expert software enabled the creation of statistical and numerical optimization models, as well as simulated regression models, to find the optimal tristimulus color values with minimal color variance (***dE****). The three examined parameters significantly affected the color parameters ***dL****, ***da****, and ***db****, and specific mechanical energy (SME) based on the analysis of variance (ANOVA). In addition, SME was calculated for the experimental trials. A decrease in SME was found as the FRate increased. The collected data were analyzed to determine pigment dispersion using scanning electron microscopy (SEM) as well as micro-CT (MCT) scanner images. Regarding the BBD, the processing conditions revealed a minimum deviation of 0.26 but a maximum design desirability appeal of 87%. The three-level full-factorial design (3LFFD) revealed a maximum desirability of 77% and a minimum acceptable color variation (***dE****) of 0.25. Therefore, BBD had a marginally superior performance. These results demonstrate that the processing parameters have a significant impact on the output quality, including reducing variation, improving color consistency, minimizing waste, and promoting sustainable production. This study found that both sets of process parameters were statistically significant after comparing the two designs. However, BBD is the preferred design for the selection needed and offers better outcomes in future experiments.

## 1. Introduction

Creating plastic with a commercially viable color necessitates the use of more than one pigment; however, attaining the desired hue on the initial attempt is a significant hurdle. Several studies have examined how different process variables affect the final color of polymer compounding [1,2]. Spectrophotometers are indispensable for quality assurance, color measurement, and the numerical comparison of color changes [3]. For the polycarbonate grade, limitations for ***dE*** are established at or below 1.0, and for ***dL****, ***da****, and ***db****, these limitations are at or below 0.6. However, it is noted that, in most cases, customers determine a suitable tolerance limit [4]. The “***dE****” represents the dispersion in (***b****), (***L****), and (***a****), as follows:(1)$$dE=\sqrt{(∆{L)}^{2}+({∆a)}^{2}+{\left(∆b\right)}^{2}}$$

In place of utilizing color values that are absolute, color differences concerning target values regarding ***dL****, ***da****, and ***db**** are used. This research examines the color difference in the CIELAB color space, using the total change in color ***dE**** [5]. This study identifies the optimal methods for utilizing statistical tools, such as DOE and processing parameters, to enhance the quality and accuracy of color in colored polymer products. It accomplishes this by extrusion, blending, and color testing, as well as by studying their rheology and response surface methodology (RSM) and examining their surface morphology and microstructural characteristics [6,7,8]. These pigments, typically of different colors, reflect incident light at various depths, resulting in the desired color being perceived as the combined effect of reflection [9,10]. Color deviations can result from a variety of factors, including differences in formulations or pigment dispersions, as well as the effects of processing parameters on material performance. This study offers a unique opportunity to examine the results of these variations in color combinations when compounding. According to the CIE, ***L****, ***a****, and ***b**** color values are obtained using a spectrophotometer [11,12].

Adding additives to polymeric components may lead to unforeseen effects on their viscosity, mechanical processing, properties, and visual appearance [13,14]. When the optimal combination of FRate, T, and Sp values is achieved under specified processing conditions, a slight deviation from the expected color output is observed [15,16,17,18]. The use of RSM has been shown to improve the performance of polycarbonate nanocomposites by enabling an examination of how key factors affecting material properties interact [19,20].

The color characteristics of a compounded polycarbonate grade are investigated to determine the impact of process parameters on these features. Consequently, a decision is made to construct a regression model. The color disparity is caused by a myriad of factors, and it is necessary to investigate how these factors affect the final hue [21,22]. In order to optimize process variables, various experimental designs are practical. The impact of varying compounding process factors on the surface appearance and gloss of PVC sheets is studied using a modified general factorial design of experiments (DOE) [23]. Some examples of RSM designs include the factorial design, central composite design (CCD), Box–Behnken design (BBD), and D-optimal designs [24]. Research has shown that a DOE using the BBD may determine the relationship between processing factors and the variance in the viscosity of the wood–plastic matrix [25].

As a combined array design, the Box–Behnken design (BBD) streamlines the computation of substantial interactions and reduces the number of runs needed when compared with Taguchi’s crossing array designs [26]. With this method, one can construct a quadratic model with just three levels per factor, which is quite economical in terms of runs [26,27].

The first step in efficiently evaluating a model’s elements is to set up the trials carefully. Then, a second-order polynomial is used to describe the responses [27,28]. On the other hand, polyacrylic acid–graphene–oxide, a compound with multiple applications, has its synthesis parameters optimized using the response surface methodology (RSM). Results from further analyses of variance (ANOVA) validate the models [29,30,31,32].(2)$$y=\beta_{o}+\sum_{i=1}^{k}\beta_{ij}x_{i}+\sum_{i=1}^{k}\beta_{ij}x_{i}^{2}+\sum \sum_{i<j=2}^{k}\beta_{ij}x_{i}x_{j}+\varepsilon$$

To obtain consistent color properties in plastics, this research looks at mixing two polycarbonate resins. Optimizing for speed, temperature, and FRate, this study employs DoE and RSM to see how pigment dispersion is affected. Improved production of colored polymer goods results from the use of prior industry data that address color consistency difficulties [33,34]. Some studies argue that when polycarbonate (PC) is mixed with low-density polyethylene (LDPE), it may reduce the overall viscosity. This allows the components to glide about during injection molding, therefore improving the process [35]. The study concludes by examining letdown and pigment dispersion and how they are affected by the processing temperature to growth ratio in polymer matrices. It examines thermal stability plus viscosity parameters [36].

SEM offers critical, invaluable information about polycarbonate blends’ characteristics (particularly the microstructural ones), especially for examining agglomeration and pigment dispersion. The findings demonstrate significant shear thinning exhibited by Acrylonitrile-Butadiene Styrene (ABS), while polycarbonate (PC) essentially exhibits Newtonian behavior [37,38]. Also, scanning electron microscopy (SEM) is used to find out how pigment dispersion quality affects color uniformity. It has been found that consistent dispersion reduces color variations and clumping [39].

Response surface methodology (RSM) provides a powerful statistical tool for optimizing processes with minimal experimental runs, making it particularly suitable for materials science and engineering. By capturing variable interactions and generating clear mathematical models, RSM is more cost-efficient than methods like Taguchi or full-factorial designs. RSM has been successfully used to optimize the thermal conductivity of nanocomposites, achieving a high R^2^ value of 96.23%. RSM offers clear interpretability, but it is less effective with highly nonlinear systems compared to artificial neural networks (ANNs).

The choice between RSM and ANN depends on the system’s complexity, the availability of data, and the computational resources at hand [40].

When comparing RSM with Artificial Neural Networks (ANNs) for optimizing the thermal conductivity of polymer nanocomposites, it is found that RSM encountered difficulties with nonlinear patterns. However, interpretable models are provided. In contrast, ANN requires more data and computational resources but offers higher accuracy. This study concludes that RSM is best suited for initial analysis, while ANN is more effective for complex systems. Hybrid approaches can combine the strengths of both methods [41].

The significance of this study lies in the identification of correlation between processing parameters and color outputs, thereby enabling color experts to enhance the reliability of color formulations. The goal is centered on the utilization and comparison of two designs to optimize processing parameters, with the aim of achieving minimal deviation in color properties (***dE**** < 1.0). In addition, more characterizations are conducted to assess the structure and efficiency of color dispersions. Both designs determine optimal colors and generate statistically significant models.

## 2. Materials and Methods

### 2.1. Materials

Trials are conducted on the material under investigation at the industrial plant. As detailed in Table 1, a combination of various pigments and two PC resins is utilized, with these grades’ color formulation expressed in Parts per Hundred (PPH). Resin 2 exhibits a value of 6.5 g/10 min, whereas Resin 1 records a melt-flow index (MFI) of 25 g/10 min.

### 2.2. Sample Preparation

Considering the 37:1 length-to-diameter ratio and a 25.5 mm diameter, the material extrusion process is conducted using a co-rotating (27-kW) Twin-Screw Extruder (TSE). The innovative intermeshing designs from Germany are utilized, specifically the Coperion ZSK26 model. Nine heating zones are located on the extruder’s barrel, while one is at the die. Following the extrusion of the melt, it is first cooled using cold water. Secondly, it is allowed to air-dry, and thirdly, it is pelletized. Three rectangular chips measuring 3 × 2 × 0.1 are produced from the pellets obtained from each run through the process of injection molding. Then, an American X-Rite spectrophotometer (CE: 7000A) is utilized for the measurement of (CIE ***L****, ***a****, ***b****). In this case, ***L**** = 70.04, ***a**** = 3.41, and ***b**** = 18.09 will produce the intended color result.

### 2.3. Experimental Design and Statistical Optimization

This research includes three process parameters into the experimental design, as shown in Table 2: extruder FRate, heating zone temperature, and screw speed. The eighth version of American Design Expert Software Stat-Ease, Inc., Minneapolis, MN, USA, is utilized to quantitatively assess the data and establish correlations among the variables at a 95% confidence interval through statistical analysis. To compare the two design levels (BBD and 3LFFD), Table 2 highlights the differences between the two designs, making it easier to view the parameter values at each level.

The temperature values remain identical across all three stages for the 3LFFD and BBD designs. In terms of speed, 3LFFD starts at a higher initial rpm (700 vs. 650), while both designs share the same middle level (750 rpm). However, they differ slightly at the highest level (800 rpm for 3LFFD vs. 850 rpm for BBD). For flow rate, 3LFFD consistently uses higher values across all levels, starting at 20 kg/h compared to BBD’s 11 kg/h, and continuing with higher values at the middle and high levels.

To make sure the data stayed true to the target color, they were optimized numerically. This experiment uses a ratio of 100:0.86 to combine the resins and additives, as shown in Figure 1. A super floater batch blender blends them according to weight ratios to create a more consistent mixture [42,43]. In the experimental design, the three levels denoted by the code (−1, 0, and +1) for each element, as presented in Table 2, assessed the same three process parameters. Both experimental designs took into account the utilization level in addition to the extruder FR, speed, and temperature, as indicated in Table 2.

### 2.4. Compounding Plastic Grade

The batches are prepared using additives and pigments. The transparent PC-grade material that is created is utilized to study the influence of processing parameters and formulation on changes in the final appearance of color and characteristics. The results are examined, and the roles played by the dispersion morphologies of the three processing parameters are analyzed, leading to a deeper understanding of the factors that create color disparities. Figure 2 displays a simplified flow diagram that illustrates the steps involved in the compounding process. This illustration clarifies the parameters. It shows their impact on the final product’s color and characteristics. It helps to understand how processing adjustments lead to different outcomes.

### 2.5. The Effects of Processing Parameter Interactions

The production of accurate color through injection requires an appropriate operational procedure, as any change in processing parameters can affect color variations.

The present study employs a systematic approach to manipulate operational factors and examine their impact on pigmentation. This research independently modifies (the speed, the rate of flow (FRate) and temperature) processing parameters across three separate stages. It is noted that, with all other variables held constant, a significant correlation exists between coloration and the processing parameters. The experiments are conducted in the following manner: The recommended processing conditions include a steady speed (750 rpm) and an FRate (25 kg/h), along with three distinct temperatures (280 °C, 255 °C, and 230 °C). Then, the same processing is repeated for the speed and the rate of flow (FRate). This study utilizes two designs of the response surface method (RSM).

#### 2.5.1. Three-Level Full-Factorial Design

A total of twenty-seven independent experimental runs make up the initial design of experiments (DOE), each with an additional five center points included to detect curvature in the response surface, detect nonlinearity in the responses, and estimate the experimental error. This results in 32 treatments, as illustrated in Figure 3a.

#### 2.5.2. Box–Behnken Design (BBD)

Figure 3b display the second design of experiments (DOE). The second design of experiments (DOE) includes 12 experimental runs and 5 additional center points to accommodate various processing parameters, resulting in a total of 17 treatments. The experimental design considers three process parameters: the extruder’s screw speed, FRate as well as the heating zones’ temperature, the extruder’s screw speed as well as FRate, with Table 2 displaying the levels used. This research varied the parameters of 12 different treatments. To assess the impact on color, five additional center points are added for the BBD response method. Subsequently, five additional center points are added to detect nonlinearity in the responses and to estimate the experimental error. The percentage load during the experimental runs is also recorded to facilitate the calculation of the specific mechanical energy (SME) using Equation (3) [44].(3)$$SME=\frac{n\cdot P\cdot O}{n_{m}Q}KWH/kg$$
where: P: power (kW); n_m_: Max. screw rotations (rpm); O: load (%); n: screw rotations (rpm); Q: FRate (kg/h).

The extruder’s screw speed, FRate as well as the heating zones’ temperatures were examined. A control study was conducted for examining the screw speed, FRate as well as temperature’s effects on color, as these factors directly influence it. This research individually controls three processing parameters at different stages. The FR and temperature are fixed at middle values, and speeds of 700, 750, and 800 rpm are subsequently selected. The same procedure is subsequently repeated for the flow rate (FRate) and temperature. A significant correlation between color-processing variables is confirmed through observation [45].

Various scientific design techniques are employed in this study to minimize material rejects during the initial batch of color evaluations. The primary objective is to investigate the interactions between FRate, temperature, screw speed, and trismillus color variants, as well as the effects of various processing conditions. In order to maximize the effect of the processing parameters, two experimental designs are employed, followed by a statistical comparison of both designs. The statistical procedures study consists of three steps. The interaction between color output, FR, temperature, and screw speed is examined. The two designs are compared using ANOVA, overlay plots, and desirability graphs. The specific mechanical energy process is examined in detail in this research. The colors in the experiments are also measured. MCT and SEM microscopes are utilized to examine the morphology of PC compounds. The shape, dispersion quality, and color consistency, along with the correlation of PC and PC compound quality of pigments, are also examined.

### 2.6. Morphological Analysis

#### 2.6.1. SEM Image Characterization Analysis

The observations in polycarbonate composites aim to describe the dispersion of the pigment. Also, these observations aim to determine the impact of various processing parameters on these samples. This study extended the focus to develop a quantitative methodology using SEM, combined with 3D-X-ray MCT scanning. To investigate the microstructures of PC-grade materials, the study uses the Scanning Electron Microscope (SEM) Model Joel (5500 LV) (JEOL Ltd., Tokyo, Japan). See Figure 4a,b for more information. Figure 4 shows that the compounded grade contains both primary particles and agglomerates. As evidenced by the interaction of FRate (B) and screw speed (C), this allows for adjustments in processing parameters, such as FRate and screw speed. Each sample is treated using different processing procedures and levels. Significant agglomerated pigment is revealed in SEM micrographs under inadequate processing conditions. It is observed that increased agglomeration occurred at lower processing temperatures and velocities. However, it might deteriorate and agglomerate at higher screw speeds.

Another SEM, Model JSM-600 (JEOL Ltd., Tokyo, Japan), see Figure 5, is used with a 20 KV acceleration voltage, a magnification of 3000×, and a working distance of 15 mm to characterize raw pigment without coating. This helps verify the presence of agglomerates in the (red, yellow, black and white) pigments and the primary particles’ existence in the vicinity of <100 nm. To ensure accurate dispersion of the additives, use high-quality equipment. When the production of specific grades of polycarbonate fails to properly blend or disperse additives such as colors, fillers, or reinforcements, agglomeration can occur.

This can cause the additives to agglomerate or cluster in the polymer matrix. Incompatible blends, such as polycarbonate grades or other polymers, can lead to phase separation and agglomeration. At low temperatures, it is observed that the processing material does not melt uniformly, resulting in visible agglomerates and inadequate mixing. Thus, excessive temperatures may cause additive deterioration or thermal instability, enhancing aggregate formation.

The size of the red pigment particles is depicted in Figure 5, measuring approximately 0.1 μm. Concerns have been raised regarding the adequacy of the SEM’s resolution for observing the finer details of the pigment particles at this level of magnification. This may result in the oversight of microscopic clumps. Research that studied uncoated pigments suggests that coatings may yield more accurate insights into particle interactions and behavior in diverse environments. However, it is revealed that the pigments are agglomerates. Also, it is determined that dispersing them is critical to improving the color quality of polycarbonate. Figure 5 shows a micrograph of the primary particles, illustrating that the pigments comprise agglomerates [7,8].

#### 2.6.2. Micro-CT Scanner (MCT) Image Characterization Analysis

A μCT scanner provides more precise information regarding material properties at a smaller scale than a digital optical microscope (DOM). This study assumes that the particles have a spherical form. This is because calculating the size of particles with non-standard shapes is a difficult task. This study uses the area of a circle as a metric for each particle. The particles are a mixture of round and non-round shapes. The circular sizes of these pigment agglomerations vary from 1 to 10 (µm) [7,8].

This study’s primary focus shifted to developing a quantitative method that utilizes DOM observations, SEM, a particle size analyzer (PSA), and an MCT scanner to demonstrate the distribution of pigments in polycarbonate composites. However, an overreliance on technological instruments may impede the development of fresh, alternative ways to yield valuable data on pigment behavior in various settings.

In this research, the high-resolution SkyScan 1172 µCT scanner allows us to capture images in three dimensions. A 10-megapixel X-ray detector is used with a 4000 × 2300 resolution to capture detailed pictures, while our system is operating at 32 kv and 187 µA. Acquiring data using a 12-bit CCD camera allowed for accurate structure characterization of the sample. Figure 6 displays the micrograph pictures, featuring improved settings and a consistent distribution of compounded PC pigment samples.

The pigment aggregation at 250× is visualized using a CT scanner; various means of characterization display morphological micrographs, aggregation, and pigment distribution. In conclusion, the results show that the µCT scanning method is dependable for examining compounded polycarbonate pigments. In addition to enabling precise vision, this method also helps comprehend how the pigments interact with the polycarbonate matrix. Building on these findings, future research will explore how different processing conditions in diverse applications affect pigment behavior and performance.

## 3. Discussion and Results

This study examines two model designs: 3LFF and BBD. Initially, the ANOVA results should be presented to develop a valid and significant model.

### 3.1. Analysis of Variance: ANOVA

An ANOVA is conducted to determine how interactions occurred and to adjust the color-processing parameters. The investigation into how different characteristics affect ***dL****, ***da****, ***db****, and SME is conducted thoroughly using the design of experiments (DOE) approach. A linear model is initiated in the experiment, with additional terms being progressively incorporated, whether linear or quadratic, based on the requirements; sequential F-tests are conducted [46]. The F-statistic for each model type is calculated in the experiment to identify the highest-order significant terms [30]. A similar methodology is followed across all tristimulus values, with only the significant terms identified by F ≤ 0.05 and probability values ≤ 0.1 included.

The result of ANOVA (for the sequential model sum of squares) in Table 3 includes ***dL****, ***da****, ***db****, and SME. The highest-order model with significant terms is the quadratic model, as indicated by a *p*-value of less than 0.05. As a result, these models accurately characterize ***dL****, ***da****, and ***db****, as well as SME responses. The adjusted R^2^ values provided in Table 3 supported this observation. Conversely, keep in mind that just because quadratic models are standard, it does not mean they are helpful in real-world contexts. Furthermore, the Prob > F values suggest statistical significance.

The components usually explain only 72% of the variation in ***db****, while the variation in ***da****, ***dL****, and SME equals 90%, according to the R^2^ in BBD. The components in 3LFFD typically account for about 78% of the difference in ***db****, ***da****, and ***dL****, as evidenced by R^2^. CIE Lab color difference metrics, where: ***da**** = change in red–green chromaticity (**Δ*a****), ***db**** = change in yellow–blue chromaticity (**Δ*b****), ***dL**** = change in lightness (**Δ*L****). However, the figure drops to 75%. There is no explanation for the residual variability in the responses. The term “noise” may be used to describe these unforeseen variations. Reasonable agreement is apparent between the “Predicted R^2^” and the “Adjacent R^2^”. “Adequate precision” is a metric that measures the ratio of noise to signal; ideal values surpass four. The ratio > 4 suggests that the model may successfully traverse the design space. To compare the pure and residual errors, the “lack of fit” test is employed. It is found that the result is not statistically significant due to the *p*-value of “lack of fit” being greater than 0.05. The F-value indicates statistical significance compared to noise, while the modest *p*-value (<0.05) indicates that the model terms are meaningful. The R^2^-value and adjusted R^2^-value illustrate that values closer to 1 suggest a better match [30].

### 3.2. Regression Models for Trismilus Color and SME

Table 4 displays the (ANOVA) results, which enable the generation of multiple-linear-regression models for ***dL****, ***da****, ***db****, and SME prediction. Coefficient values demonstrate the extent to which process variables (FRate, Sp, and T) impacted responses; polynomial equations mathematically depict these variables and their interactions. Table 5 and Table 6 display the tiny variations between the predicted and actual color responses, indicating a significant agreement.

### 3.3. Interaction of Feed Rates (FRate) and Speed for Tristimulus Color Values

To compare the two designs, this study employs a three-level factorial design (3LFFD) and a Box-Behnken design (BBD) for the processing parameters. Processing parameters’ influence, particularly Sp and FRate, on the tristimulus color values of polymer color chip samples is the primary focus of this comparison and in-depth analysis. This interaction is critical for assuring the final color consistency of polymer–plastic products during production. This study aims to investigate these two variables to determine the optimal processing settings for achieving exact and consistent color results, utilizing tristimulus color values for measurement and computation.

#### 3.3.1. Interaction of Speed and FRates for ***dL****

Figure 7a illustrates the design interactions in BBD at 274.4 °C, presenting a contour graph that shows the relationship between FRate and speed for ***dL****. In contrast, at 245.2 °C, Figure 7b (3LFFD) depicts the link between speed and FRate. The FRate and speed nonlinear complex relationship is illustrated in BBD in Figure 7a. When (24.4) kg/h and (728) rpm, ***dL**** is −0.01: the global best value. In contrast, 3LFFD, Figure 7b, shows that the worldwide (***dL****) ideal value is equal to 0.0, particularly at 245.2 °C, 24.71 kg/h, and 741.2 rpm. Using a 95% confidence interval, the lowest possible value of ***dL**** is −0.11, while the maximum is 0.11.

#### 3.3.2. Interaction of Speed and FRates for ***da****

Figure 8a presents a BBD contour map of the speed–FRate interaction at 274 °C. In contrast, Figure 8b of the 3LFFD shows the speed–FRate interaction for da*, demonstrating a first-order link between speed and FRate, particularly at 245.2 °C. Figure 8a displays the BBD elliptical shapes characterized by a large radius of curvature, which results from the quadratic components of the three processing parameters.

This indicates that as FRates and speeds rise, the ***da**** value decreases in two quadrants and increases in the other two. Additionally, two quadrants exhibit positive FRate–speed interactions, while the other two exhibit negative ones. These positive interactions underscore the potential for optimized performance. Conversely, as shown in 3LFFD (Figure 8b), experimenting with different FRates and speeds assists in reaching the objective. A 95% confidence interval indicates that ***da**** = 0.15 at 24.7 kg/h at 741.2 rpm, with 0.07 being the lowest acceptable value and 0.23 being the maximum, where 95% CI low: lower 95% confidence bound of the average predicted values; 95% CI high: upper 95% confidence bound of the average predicted values; 95% PI low: lower 95% prediction bound for a future observation; 95% PI high: upper 95% prediction bound for a future observation. These findings will help further studies proceed with their experimentation, providing significant benefits in terms of productivity and efficiency.

#### 3.3.3. Interaction of Speed (Sp) and Feed Rates (FRate) for ***db****

Figure 9a in BBD illustrates the ***db**** speed–FRate relationship at 274.4 °C, demonstrating a considerable nonlinearity level, similar to ***dL****. On the other hand, Figure 9b, 3LFFD, at 245.2 °C, illustrates the speed–FRate relationship for ***db****. The behavior is highly linear, with only speed and FRate as major model variables compared to (***dL**** and ***da****) [30].

Figure 9a in BBD shows a notable finding for ***db**** (approaching zero), while maintaining a temperature of 274.4 °C in both cases, at around 23 kg/h FRates and the highest speeds, and around an FRate of 15 kg/h, while the speeds are the lowest. ***Db**** equals 0.19 (as the anticipated value) at the global optimum. On the other hand, in Figure 9b, 3LFFD, notably ***db****, approaches zero at lower speeds and FRate values around 20 kg/h, as well as at higher values around 24 kg/h, all while maintaining a temperature of 245.2 °C. At the global optimum, the anticipated value of ***db**** is −0.19, with the permissible lowest and maximum values at a 95% confidence interval being −0.33 and −0.06, respectively [30].

#### 3.3.4. Speed–FRate Interaction for SME

From an energy consumption perspective, the specific mechanical energy is crucial and is maintained at a minimum for economic reasons. The specific mechanical energy depends mostly on FRate and speed [46]. According to Figure 10, a contour plot shows the FRate–speed relationship for a given amount of mechanical energy. Figure 10 illustrates that an increase in FRate results in a decrease in specific mechanical energy, whereas an increase in speed has a minimal effect on it.

#### 3.3.5. Desirability Processing Interactions for RSM

Some experts argue that a multi-criteria decision-making strategy is necessary due to the significant influence of these parameters (FRate (C), speed (B), and temperature (A)) on the outcomes [47]. However, this strategy may complicate our analysis. This study employs a total-desirability function, “d”, to determine the optimal values for these three factors in each solution [47]. The desirability function shows the desired ranges for each response (di) and provides an overall quality score. It makes it simple to compare many responses in order to choose the most desirable characteristics [30].

The measured responses are translated into a dimensionless desirability scale by calculating scales from d = 0 (an entirely undesired reaction) to d = 1 (a fully wanted answer). The best outcomes are achieved when all criteria are combined optimally globally. Assuming an FRate of 24.44 kg/h, Figure 11a depicts a three-dimensional plot of global desirability. Our findings suggest that at 24.44 kg/h, 274.23 °C, and 728.38 rpm, the combined desirability reaches a peak of 87%. At roughly 255 °C for temperature and 775 rpm for speed, the desired peaks reach a maximum value of around 0.87. The statistics also demonstrate that the desirability peaks are highest in the ranges’ centers and lowest at their edges. The trend illustrates a correlation between departures from these temperature and speed thresholds and proportional decreases in desirability.

Figure 11b illustrates a three-dimensional plot of global desirability (D) with a feed rate of 24.7 kg/h. This experiment identifies the best operating parameters for achieving a maximum combined desirability of 77%: 245.2 °C, 741.2 rpm, and a FRate equal to 24.7 kg/h. The remaining experimental domain displays diminishing d values. For example, in Figure 11b, it is evident that higher and lower FRates and speeds have a greater impact on attractiveness than temperature does in relation to speed.

#### 3.3.6. Graphical Optimization (Overlay Plots) for RSM

In Figure 12a, BBD displays an overlay plot of response contours that illustrate speed against temperature, allowing you to see the range of possible values in the illustration. To obtain the mean responses (dL*, da*, db*) at a flow rate of 24.44 kg/h, the contours with ***db**** = 0.20 and ***da**** = 0.30 border the region, indicating the ideal speed and temperature parameters. At 274.23 °C, 728.38 rpm, and 22.44 kg/h, the best tristimulus values are achieved—***dL**** = −0.01, ***db**** = −0.19, ***da**** = 0.18, and SME = 0.389. Compared to the highest allowable deviation, ***dE**** = 1.0, the overall minimum tristimulus values’ deviation is 0.26 obtained from Equation (1), which falls within the permitted range [30].

An overlay plot of response contours with speed vs. FRate at 245.2 °C is shown in Figure 12b. It shows the possible values. For an FRate equal to 24.44 kg/h, the speed as well as the temperature operating parameters that meet the mean responses (***dL****, ***da****, ***db****) are shown by the area between the contours of ***db**** = 0.20 and ***da**** = 0.30, when (741.2 rpm, 22.7 kg/h, and 245.2 °C). The optimal tristimulus values of ***da**** = 00.15, ***dL**** = 00.0, ***db**** = −00.19, and ***dE**** = 1.0 are achieved, compared to the highest allowable deviation. The overall minimum deviation in tristimulus values of 0.25 produced by Equation (1) is considered relatively acceptable [30].

#### 3.3.7. Optimization of Processing Parameters and Desirable Color Outputs: A Design Methodology

Table 7 displays the ideal conditions, color output (**Δ*****E****), necessary processing parameters, and attractiveness. This analysis compares two design methodologies: 3LFFD and BBD.

At 274 °C, 728 rpm, and 24.4 kg/h, this study achieved the best results using the Box–Behnken design (BBD) specifications. The total number of runs completed is 17, and the desirability is 87%, indicating a superior level of optimization. Many depend on the processing parameters, especially the relationship between FRate and speed for all color values (***dL****, ***da****, and ***db****). The resultant color output (***dE****) is 0.26, based on the calculations.Identification of Important Processing Parameters using a Three-Level Factorial Design: The primary impacts, as well as a few interactions (e.g., AB, AC, BC), comprise the simpler set. The preferred settings are as follows: FRate is set at 24.4 kg/h, speed is set at 734 rpm, and temperature is set at 255.7 °C. The color output (***dE****) is 0.25. It is significantly closer to the goal color of BBD. The desirability is 77%, which is slightly lower than BBD. BBD shows superior optimization. It has a modest decrease in color accuracy, evidenced by a slightly higher ***dE**** of 0.26 and a higher attractiveness score of 87%. The three-level design brings ***dE**** somewhat closer to the goal hue (0.25). It has a lower attractiveness (77%). This means it is not as good at optimizing for balance.To sum up, the BBD proves to be more robust for overall optimization and desirability. BBD costs less because it requires fewer optimization runs. The three-level design achieves color accuracy better than BBD. It is more cost-effective due to fewer optimization cycles. BBD is more reliable. The color value difference between the two designs is a negligible 0.01. BBD has a higher desirability of 10% and operates at higher temperatures of 18.3 °C.Although there are some minor compromises in color accuracy, the BBD presents a viable option for limited-budget projects due to its lower cost and greater desirability. This potentially increases energy costs, as the processing requires higher temperatures (274 °C). Additionally, 3LFF necessitates longer runs than BBD, resulting in higher material costs.

The key significant findings of this study for the comparison between the two design models are summarized in Table 7, as shown below:Significant ANOVA overlap interaction alignments are (***dL****) A, C, BC, (***da****) BC, and (***db****); there is no overlap.The optimized difference in processing parameters is as follows:Temp: Greater energy difference with (18.3 °C).Speed: Minimal effect variation in speed (6 rpm).Feed rate: A significant independent optimization with no observed difference (zero).BBD offers a more efficient design with fewer (17) runs than 3LFFD: “The testing process demonstrated considerable material savings, requiring just 17 runs as opposed to the 32 runs commonly necessary with screw extruders”.BBD demonstrates superior optimization desirability performance with 10%.Minor color difference (***dE****) observed between the two models.

### 3.4. Quantitative Morphological Characterization Based on SEM

#### 3.4.1. Color Change as a Function of Processing Conditions

Table 8 illustrates the effect of the feed rate on the color difference (***dE****) at 20 kg/h, 25 kg/h, and 30 kg/h, while the temperature of 255 °C and speed of 750 rpm are fixed. Color differences (***dE****) show a higher color difference when the feed rate is 20 kg/h. A pigment particle in this sample shows incomplete dispersion and higher agglomerations than the other two samples. At 30 kg/h, these findings indicate that a slight reduction in the color difference appears when the feed rate is higher.

#### 3.4.2. Examine Dispersion at Variant Feed Rates

In general, the total color differences decrease when the feed rate is increased. According to this result, the 30 kg/h sample has a more evenly dispersed pigment than the 25 kg/h sample, which, in turn, is more evenly dispersed than the 20 kg/h sample. Figure 13 shows that the peak of the distribution becomes narrow, e.g., the feed rate of 30 kg/h, at peak 53.8% and the particle sizes lie at approximately 0.8 microns, in comparison to 52.4% of particles lying at 0.8 at 25 kg/h, and 47.8% of particles lying at 0.97 µm at 20 kg/h. The sample in this study is investigated, and it is found that the particle size distribution graphs correlate well with the feed rates. In addition, the image and particle size distribution analysis findings agree with differences in color measurement (see Table 9).

#### 3.4.3. Effect of Feed Rate on Particle Morphology and Agglomeration

The 30/70% PC color-grade sample, processed at three different levels, is analyzed using X-ray microtomography, revealing consistent pigment distribution and spherical particle shapes. Improved dispersion is observed at higher processing parameters (temperature, feed rate, and speed), where pigment agglomeration is reduced, and distribution peaks become narrower. Proper processing enhances uniform pigment flow, dispersion, and color stability, as shown in supporting graphs in Figure 14(1) [14,16,33]. Figure 14(1) shows micro-CT images illustrating the pigment distribution within the blend processed at 750 rpm, 255 °C, and a flow rate of 25 kg/h. The experiment shows that the dispersion stays at a steady level, even with variations in processing parameters. When the processing conditions remain consistent, variations in the shape and distribution of pigment agglomeration, as illustrated in Figure 14(2), show pigments with a predominantly spherical morphology. It is consistently perceived that a certain degree of agglomeration and improvement in color uniformity occurred as the feed rate increased.

In conclusion, optimizing processing parameters is essential for PC mix visual quality improvements because improved pigment dispersion increases color consistency and reduces agglomeration. In summary, agglomeration affects color quality. Less agglomeration means more consistent color dispersion, ultimately improving the color performance.

## 4. Conclusions

Pigment dispersion determines polymer color quality, influenced by mixing, compatibility, and temperature. This study employed a full-factorial (3LFFD) and Box–Behnken design (BBD) approach using Design-Expert software to optimize the feed rate, screw speed, and temperature. ANOVA revealed that these parameters have a significant effect on color values (***dL****, ***da****, ***db****, ***dE****). BBD achieved optimal results at 274.23 °C, 728.38 rpm, and 24.44 kg/h with 87% desirability and ***dE**** = 0.26. While 3LFFD performed slightly better in color accuracy (***dE**** = 0.25), it had lower desirability (77%) and required more experimental runs. BBD proved more efficient, requiring only 17 runs. Feed rate showed a significant independent influence in both models. Increased feed rates improved dispersion and reduced color variation. The feed rate of the 30 kg/h sample had a 53.8% yield, with the highest percentage of particles at a low average particle size of 0.8 µm. Also, it had a spherical shape and the lowest color deviation. Both designs ensured color consistency, energy efficiency, and reduced agglomeration. However, BBD offered a better cost-performance balance.

The key findings of this study highlight that both design models—BBD and 3LFFD—showed significant ANOVA overlap, particularly in tristimulus color parameters ***dL**** and ***da****, as well as the interaction between screw speed (B) and feed rate (C). This prompted further investigation into how the feed rate affects pigment dispersion. The feed rate emerged as a consistently significant independent factor, with no notable differences in its impact between the two models. BBD proved more efficient, requiring fewer experimental runs, about 53% compared to 3LFFD. Also, BBD demonstrated a 10% improvement in optimization desirability. In conclusion, optimized processing and pigment dispersion, integrated with statistical and numerical techniques, can accomplish accurate tristimulus values with significant minimal color variance (***dE****).

## Figures and Tables

**Figure 1 polymers-17-01719-f001:**
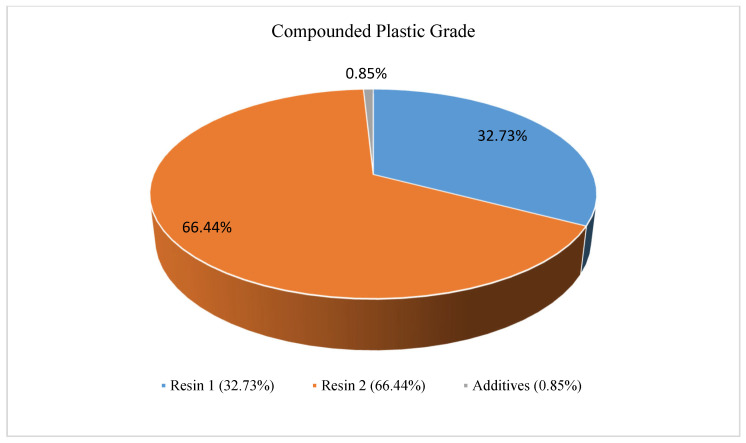
Typical compounding plastic color grade of polycarbonate.

**Figure 2 polymers-17-01719-f002:**
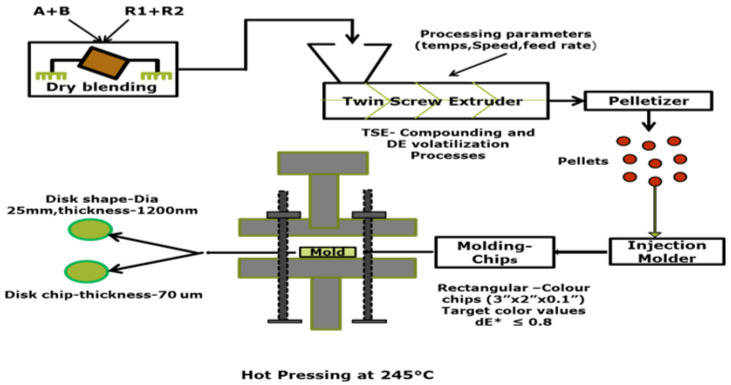
Schematic diagrams of the process methods of plastic.

**Figure 3 polymers-17-01719-f003:**
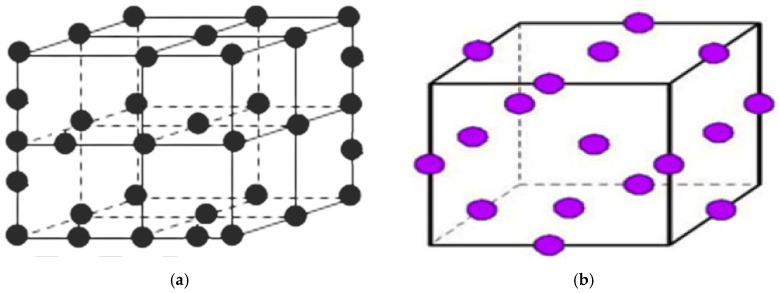
(**a**) Three-level full-factorial design (32 runs). (**b**) Box–Behnken design (17 runs).

**Figure 4 polymers-17-01719-f004:**
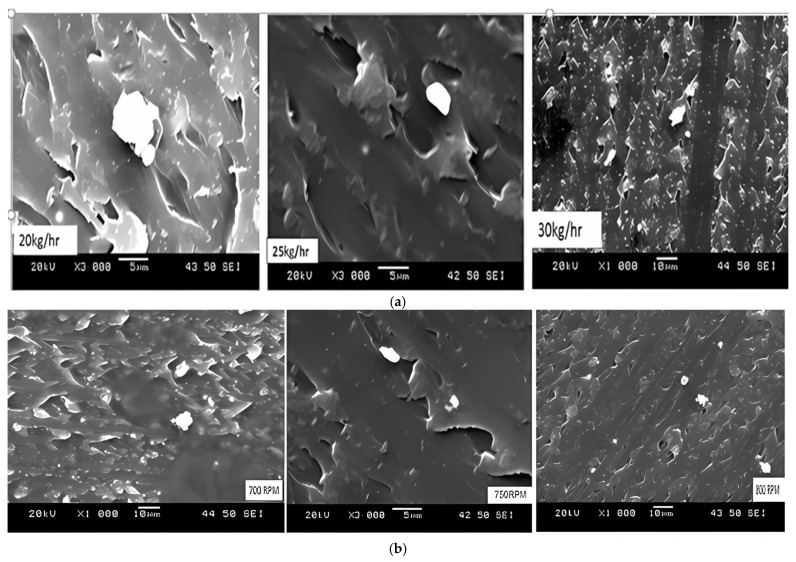
(**a**). SEM agglomeration micrographs for variation in feed rate; (**b**). SEM agglomeration micrographs for variation in speed.

**Figure 5 polymers-17-01719-f005:**
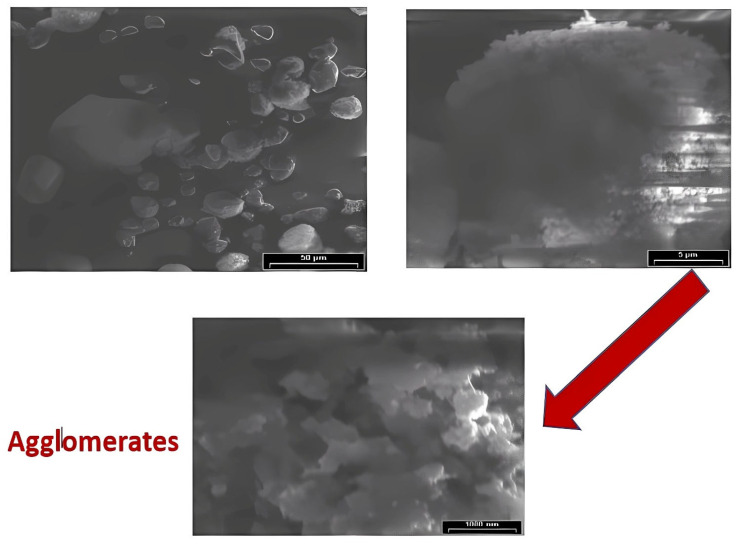
SEM micrograph of raw pigments—red, 3000× (agglomerations).

**Figure 6 polymers-17-01719-f006:**
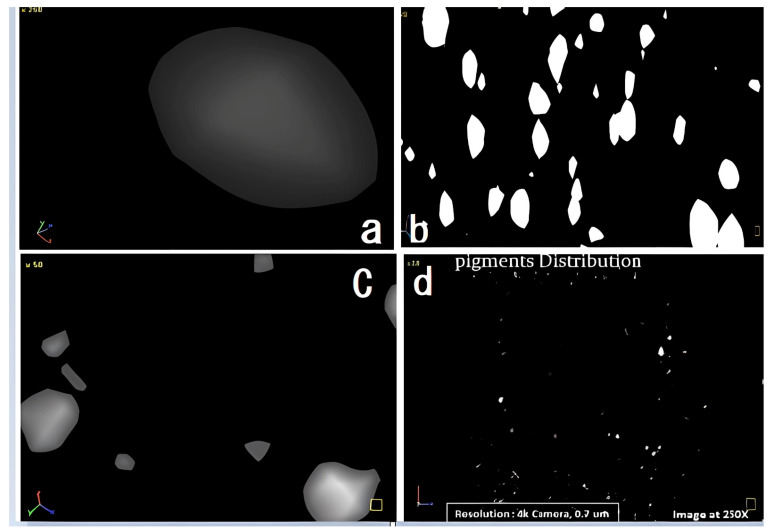
Micro-CT micrographs of PC grade at center point, image at 250×: (**a**) spherical particle shape, (**b**,**c**) agglomerations, (**d**) pigment distribution.

**Figure 7 polymers-17-01719-f007:**
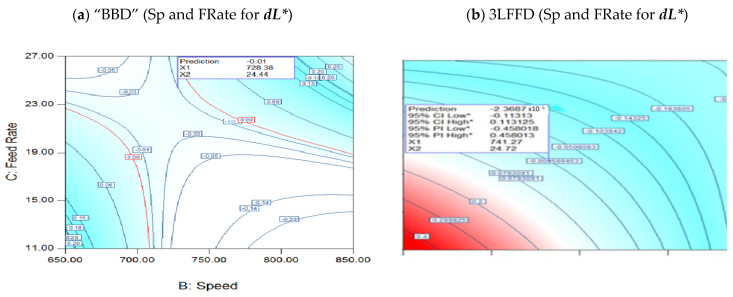
(**a**) Interaction speed and FRate for BBD at (**a**) 274.23 °C for ***dL****, (**b**) 3LFFD at 245.2 °C for ***dL****. Asterisk (*) denotes values within a 95% confidence or prediction interval, based on model predictions assuming normally distributed residuals.

**Figure 8 polymers-17-01719-f008:**
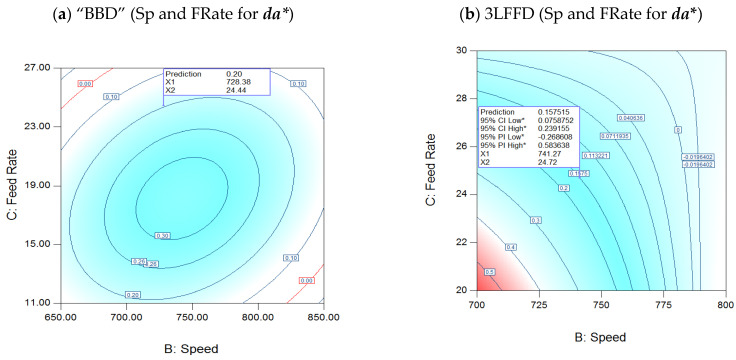
(**a**): Interaction speed (rpm) and FRate (kg/h) for BBD at 274.23 °C for (***da****), (**b**) 3LFFD at 245.2 °C for (***da****). Asterisk (*) indicates that the value is part of a 95% confidence or prediction interval. These intervals are based on model predictions assuming normally distributed residuals.

**Figure 9 polymers-17-01719-f009:**
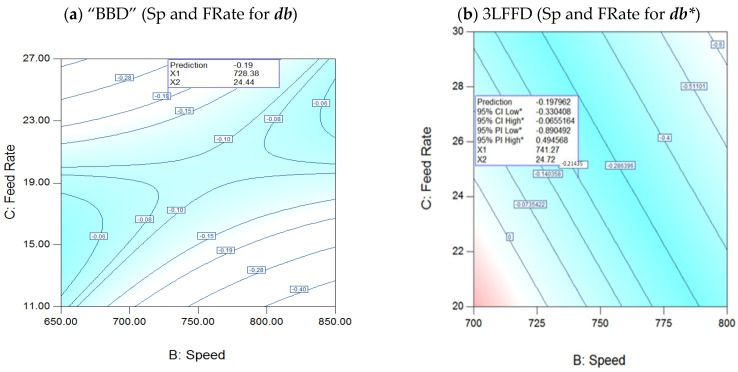
(**a**): Interaction speed (rpm) and FRate (kg/h) for BBD at 274.4 °C for ***db****, (**b**): 3LFFD at 245.2 °C for ***db****.

**Figure 10 polymers-17-01719-f010:**
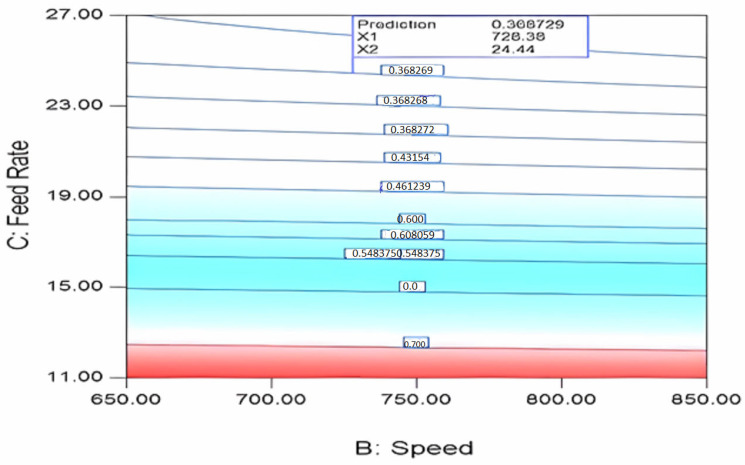
‘SME’ interaction speed (rpm) and FRate (kg/h) plot: at 274.23 °C and 728.38 rpm.

**Figure 11 polymers-17-01719-f011:**
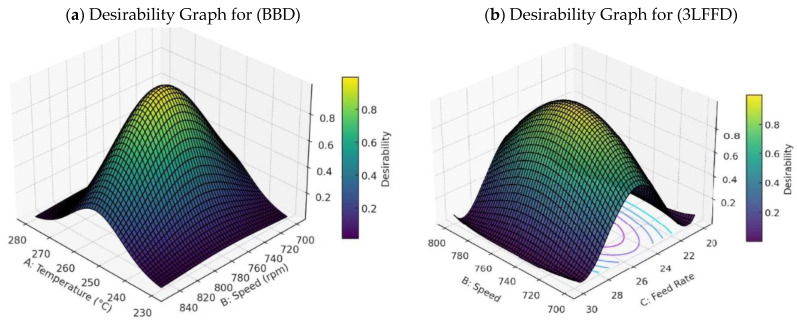
(**a**) Desirability interaction at 24.44 kg/h; (**b**) desirability graph at 245.2 °C temp.

**Figure 12 polymers-17-01719-f012:**
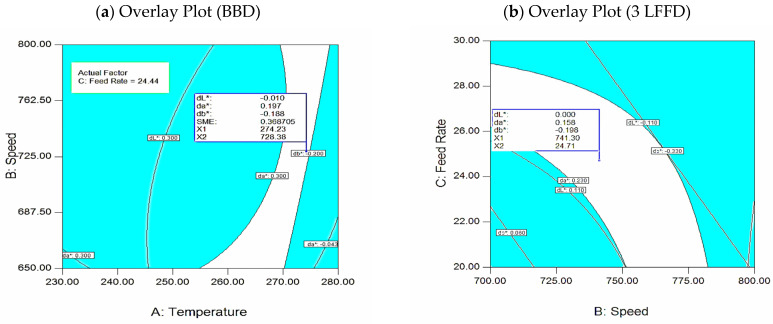
(**a**): Temp–speed contour (FRate = 24.44 kg/h), (**b**): speed–FRate contour plot: at 245.2 °C.

**Figure 13 polymers-17-01719-f013:**
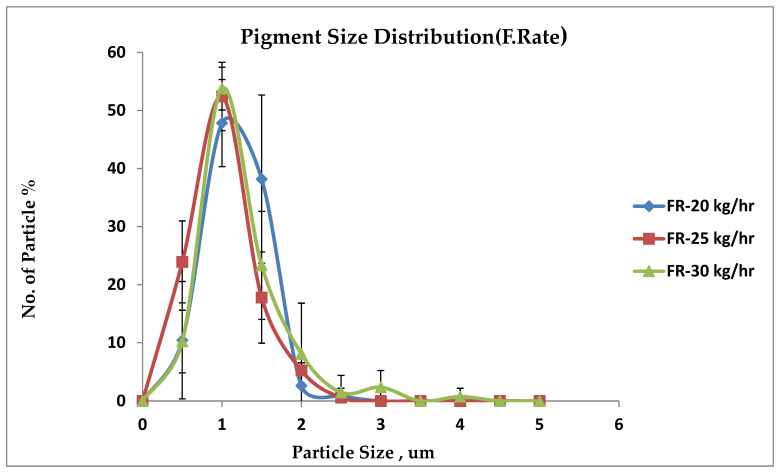
Pigment size distribution for feed rate.

**Figure 14 polymers-17-01719-f014:**
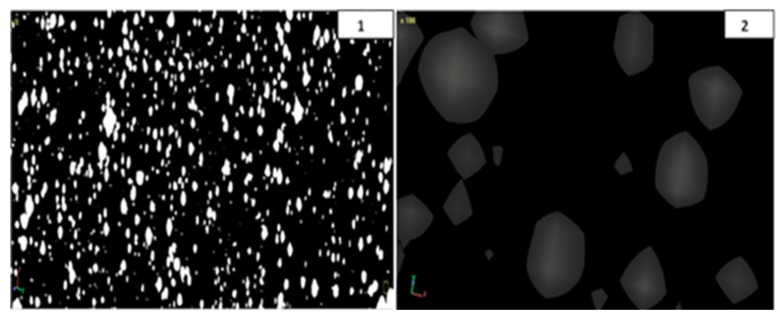
Micro-CT analysis of pigment agglomeration ((**1**) distribution, (**2**) shape).

**Table 1 polymers-17-01719-t001:** Various pigments and 2 PC resins/color formulation.

(S)	(Type)	(PerHundred)
1	Resin1	32.73
2	Resin2	66.44
3	PigmentA	00.20
4	PigmentB	00.05
5	PigmentC	00.0004
6	PigmentD	00.0016
7	PigmentE	00.0710

**Table 2 polymers-17-01719-t002:** Experimental design level and parameters for BBD and 3LFFD.

Factors	Units	3LFFD (−1)	3LFFD (0)	3LFFD (+1)	BBD (−1)	BBD (0)	BBD (+1)
Temp	°C	230	255	280	230	255	280
Speed	rpm	700	750	800	650	750	850
Flow Rate	kg/h	20	25	30	11	19	27

**Table 3 polymers-17-01719-t003:** ANOVA testing the hypothesis of RSM used with ***da****, ***db****, ***dL****, and SME.

(Resp.)		(Signif. Terms)	(R^2^)	(Pred. R^2^)	(Adj. R^2^)	(Adeq. Prec.)
BBD	(***dL****)	BC, B^2^, C, A	0.940	0.840	0.910	17.420
3LFFD		C, B, A, AB, BC, AC	0.780	0.380	0.550	0.550
BBD	(***da****)	A, C^2^, B^2^, A^2^, AC, BC	0.980	0.890	0.970	27.80
3LFFD		C, B, BC	0.750	0.240	0.390	8.530
BBD	(***db****)	A, BC, C^2^, A^2^,	0.720	0.400	0.560	5.620
3LFFD		B, C	0.750	0.280	0.300	8.610
SME		A, B, C, A^2^, C^2^	0.990	0.970	0.930	106.0

A = Temperature, B = Speed, C = FRate. (Note: The asterisk (*) is part of the standard nomenclature in color science to distinguish these parameters from non-standardized values.)

**Table 4 polymers-17-01719-t004:** (***da****, ***db****, ***dL****, and SME) Regression model.

(Response)		(Regression Model)
(***dL****)	(BBD)	+12.34563 − 0.011717 × T − 0.018803 × Sp − 0.22115 × FRate 3.09375 × 10^−4^ × Sp × FRate + 8.38889 × 10^−6^ × Speed^2^
	(3LFFD)	+63.86390 − 0.19647 × T − 0.065085 × Sp − 0.99472 × FRate 1.84353 × 10^−4^ × T × Sp + 1.96624 × 10^−3^ × T × FRate + 6.39611 × 10^−4^ × Sp × FRate
(***da****)	(BBD)	−34.33712 + 0.20508 × T + 0.024262 × Sp + 0.069289 × FRate − 2.16667 × 10^−4^ × T × FRate + 1.20833 × 10^−4^ × Sp × FRate − 4.07867 × 10^−4^ × T^2^ − 1.78250 × 10^−5^ × Sp^2^ − 2.74609 × 10^−3^ × FRate^2^
	(3LFFD)	+14.59778 − 0.018496 × Sp − 0.47296 × FRate + 5.98224 × 10^−4^ × Sp × FRate
(***db****)	(BBD)	−19.24168 + 0.18004 × T − 5.00208 × 10^−3^ × Sp − 0.077943 × FRate 2.52083 × 10^−4^ × Sp × FRate − 3.66316 × 10^−4^ × T^2^ − 2.86116 × 10^−3^ × FRate^2^
	(3LFFD)	+4.08697 − 4.78866 × 10^−3^ × Sp − 0.029746 × FRate
(SME)		+2.72593 − 9.00716 × 10^−3^ × T − 6.00329 × 10^−5^ × Sp − 0.077916 × FRate 1.64940 × 10^−5^ × T^2^ + 1.37360 × 10^−3^ × FRate^2^

**Table 5 polymers-17-01719-t005:** Actual and predicted values of **Δ*L****, **Δ*a****, and **Δ*b**** for (3LFFD).

(Run)	(Δ*L**)		(Δ*a**)		(Δ*b**)		Residuals (Actual-Predicted)		
	(Actual)	(Pred)	(Actual)	(Pred)	(Actual)	(Pred)	(Δ*L**)	(Δ*a**)	(Δ*b**)
1	−0.520	−0.50	00.61	00.57	00.31	00.14	−00.02	00.04	00.17
2	00.16	0.093	00.23	00.12	−00.17	−00.25	0.067	00.11	00.08
3	−0.59	−0.57	−0.3	−0.087	−0.76	−0.34	−0.02	−0.213	−0.42
4	0.11	−0.2	0.29	0.003	−0.22	−0.4	0.31	0.287	0.18
5	0.01	−0.24	−0.16	−0.059	−0.16	−0.49	0.25	−0.101	0.33
7	−0.65	−0.36	−0.29	0.12	−0.77	−0.25	−0.29	−0.41	−0.52
8	0.14	−0.029	0.34	0.3	−0.11	0.0081	0.169	0.04	−0.1181
9	−00.53	−00.47	0.027	0.024	−00.38	−00.16	−00.06	0.003	−00.22
10	0.087	−0.18	0.14	0.03	−0.18	−0.4	0.267	0.11	0.22
11	−0.21	−0.087	0.077	−0.087	−0.56	−0.34	−0.123	0.164	−0.22
12	−0.44	−0.54	0.65	0.24	0.37	−0.099	0.1	0.41	0.469
13	0.25	0.41	0.27	0.24	−0.14	−0.099	−0.16	0.03	−0.041
14	−00.55	−00.23	−00.15	−0.059	−00.55	−00.49	−00.32	−0.091	−00.06
15	−00.71	−0.20	−00.05	0.031	−00.93	−00.40	−00.51	−0.081	−00.53
16	−0.53	−0.4	−0.12	−0.031	−0.84	−0.64	−0.13	−0.089	−0.2
18	−0.34	−0.26	−0.22	0.024	−0.35	−0.16	−0.08	−0.244	−0.19
19	0.037	−0.065	0.29	0.24	−0.22	−0.099	0.102	0.05	−0.121
20	−0.48	−0.49	0.65	0.3	0.37	0.0087	0.01	0.35	0.3613
21	−00.20	−00.33	−0.017	−0.087	−00.33	−00.34	00.13	00.07	00.01
22	00.43	00.43	00.32	00.30	0.013	0.008	0.00	00.02	0.005
23	0.027	−0.13	0.24	0.12	−0.24	−0.25	0.157	0.12	0.01
24	0.087	0.2	0.24	0.57	−0.16	0.14	−0.113	−0.33	−0.3
27	−0.04	−0.13	0.05	0.12	−0.31	−0.25	0.09	−0.07	−0.06
28	−0.15	−0.13	0.12	0.12	−0.013	−0.25	−0.02	0	0.237
29	−0.19	−0.13	0.093	0.12	−0.083	−0.25	−0.06	−0.027	0.167
30	−0.02	−0.13	−0.087	0.12	−0.02	−0.25	0.11	−0.207	0.23
31	−0.1	−0.13	−0.09	0.12	−0.06	−0.25	0.03	−0.21	0.19
32	−0.033	−0.15	0.13	−0.031	−0.32	−0.64	0.117	0.161	0.32

**Table 6 polymers-17-01719-t006:** Actual and predicted values of **Δ*****L****, ***Δa****, ***Δb****, and SME, for (BBD).

Run	(Δ*L**)		(Δ*a**)		(Δ*b**)		(SME)		Residuals (Actual-Predicted)			
	Actual	Pred.	Actual	Pred	Actual	Pred	Actual	Pred	*dL**	*da**	*db**	SME
1	−0.12	−0.062	−0.017	−0.02	−0.36	−0.24	0.46	0.45	−0.06	0.006	−0.12	0.01
2	−0.26	−0.2	0.07	0.038	−0.27	−0.43	0.75	0.76	−0.06	0.032	0.16	−0.01
3	0.27	0.18	0.63	0.6	0.39	0.18	0.47	0.45	0.09	0.03	0.21	0.02
4	−0.037	0.006	−0.043	0.012	−0.34	−0.2	0.49	0.48	−0.04	−0.055	−0.14	0.01
5	0.006	−0.025	−0.027	−0.04	−0.29	−0.39	0.34	0.35	0.031	0.017	0.1	−0.01
6	0.3	0.18	0.65	0.6	0.39	0.18	0.47	0.5	0.12	0.05	0.21	−0.03
7	0.12	0.18	0.58	0.6	−0.003	0.18	0.47	0.47	−0.06	−0.02	−0.183	0.0
8	0.51	0.59	0.38	0.36	0.14	0.14	0.5	0.5	−0.08	0.02	0.0	0.0
9	−0.1	−0.1	0.11	0.13	−0.31	−0.25	0.77	0.76	0.0	−0.02	−0.06	0.01
10	0.15	0.18	0.58	0.6	−0.047	0.18	0.47	0.48	−0.03	−0.02	−0.227	−0.01
11	0.59	0.52	0.36	0.33	0.11	0.094	0.49	0.47	0.07	0.03	0.016	0.02
12	0.47	0.56	0.36	0.4	−0.023	−0.05	0.38	0.37	−0.09	−0.04	0.025	0.01
13	0.2	0.18	0.57	0.6	0.37	0.18	0.47	0.46	0.02	−0.03	0.19	0.01
14	0.55	0.57	0.32	0.33	0.12	0.19	0.35	0.34	−0.02	−0.01	−0.07	0.01
15	0.36	0.39	0.29	0.3	−0.13	−0.09	0.8	0.79	−0.03	−0.01	−0.043	0.01
16	0.19	0.14	0.2	0.17	−0.22	−0.17	0.36	0.35	0.05	0.03	−0.05	0.01
17	0.53	0.469	0.37	0.36	0.15	0.202	0.77	0.78	0.061	0.01	−0.052	−0.01

**Table 7 polymers-17-01719-t007:** Optimized processing parameters and corresponding desirable color outputs.

Design Methodology	Color Difference		BBD	3 LEVEL	Model Comparison: Overlap and Variance (Δ)	Implication
Significant of Color parameters	** *dL** **		A, C, BC, B^2^	C, B, A, AB, BC, AC	BC, A, C	Significant ANOVA overlap interactions alignment in (***dL****) for A, C, BC
	** *da** **		B^2^ A^2^ C^2^, AC, A, BC	BC, B, C	BC	Both ANOVA models yield In (***da****) s with overlapping BC Interaction
	** *db** **		BC, A, A^2^ C^2^	B, C	ZERO	“No overlap in (***db****) between ANOVA models”
Optimized Parameters	Temp	°C-A	274	255.7	18.3	Greater energy difference with (18.3 °C)
	Speed	rpm-B	728	734	6	Minimal effect variation of Speed (6 rpm)
	FRate	Kg/hrC	24.4	24.4	0	Independent optimization with no observed difference”
Number of runs			17	32	15	BBD offers a more efficient design with fewer (17) runs than 3LFFD”
Desirability %			87	77	Δ%=10%	BBD demonstrates superior optimization desirability performance (%)
Color output (***dE****)			0.26	0.25	Δ*E* = 0.01	Minor color difference(***dE****) observed between the two models”

A = Temperature, B = Speed, C = Feed rate.

**Table 8 polymers-17-01719-t008:** CIELAB color differences for feed rate.

Screw Runs	FRate (Kg/h)	Temp (°C)	Speed (rpm)	Color (*dE**)	Tristimulus Color Value		
					*L**	*a**	*b**
1	FR−20	255	750	0.44	68.89	1.40	15.90
2	FR-25	255	750	0.35	68.42	1.47	15.35
3	FR-30	255	750	0.32	68.80	1.51	15.64

**Table 9 polymers-17-01719-t009:** Particle size distributions related to feed rate.

Screw Runs	FRate (kg/h)	Color Value (*dE**)	No. of Particle %	Average. Pigment Size	Implication Results
1	FR-20	0.44	47.8	0.97	FR-30 sample showed the optimal color value, the highest No. % of particles and an average low particle size of 0.84
2	FR-25	0.34	52.4	0.83	
3	FR-30	0.32	53.8	0.84	

## Data Availability

The original contributions presented in this study are included in the article; further inquiries can be directed to the corresponding author.

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
