# Peer review of "Optimization and Simulation of Extrusion Parameters in Polymer Compounding: A Comparative Study Using BBD and 3LFFD"

_polymers, 2025, doi:10.3390/polym17131719_

Round 1

Reviewer 1 Report

Comments and Suggestions for Authors

In this work, the author focused on the optimization of process parameters (speed, temperature, and feed rate) in the extrusion of polycarbonate pigment blends using two statistical approaches: Box-Behnken Design (BBD) and Three-Level Full Factorial Design (3LFFD). The study evaluates color consistency (dL*, da*, db*, dE*) and specific mechanical energy (SME) using response surface methodology (RSM), regression modeling, and experimental trials. Several issues have to be addressed, as follows:

  1. Language proofreading is required for the manuscript as it contains grammatical errors, frequent misuse of terms, and improperly written phrases. Moreover, the author needs to shift from the currently adopted conversational way of writing to an academic style to enhance professionalism. 
  2. The abstract is too lengthy and needs to be shortened to include the theme and motivation of the work, methodology, main findings, and significance of the study.
  3. The introduction is too lengthy and needs to be shortened. It is rather recommended to compare with previous research work by establishing a table.
  4. Introduce more on the RSM and its significance by comparing it with literature that uses similar or different models. Example: An approach towards optimization appraisal of thermal conductivity of magnetic thermoplastic elastomeric nanocomposites using response surface methodology
  5. Detailed characterization details must be provided under methodology.

Author Response

Ist Reviewer 

  1. Language proofreading is required for the manuscript as it contains grammatical errors, frequent misuse of terms, and improperly written phrases. Moreover, the author needs to shift from the currently adopted conversational way of writing to an academic style to enhance professionalism.

Thank you for the valuable feedback. I acknowledge the need for language proofreading and will ensure that the manuscript is thoroughly reviewed to correct grammatical errors, improve term usage, and revise any improperly written phrases. Additionally, I will revise the writing to adopt a more formal academic tone, replacing the current conversational style to enhance clarity and professionalism. These improvements will help strengthen the overall quality and readability of the manuscrip

2.The abstract is too lengthy and needs to be shortened to include the theme and motivation of the work, methodology, main findings, and significance of the study.

Thank you for your feedback. I have revised the abstract to shorten it while ensuring it includes the key elements: the theme and motivation of the work, methodology, main findings, and the significance of the study. The updated abstract is now more focused and succinct, addressing all the necessary aspects in a clear manner.

3.The introduction is too lengthy and needs to be shortened. It is rather recommended to compare with previous research work by establishing a table.

I have shortened the introduction  for more 50% as requested and will include a comparison tablewith previous research work to enhance clarity and conciseness.

In terms of the revision, I have removed /added sections in the introduction that:

In terms of the revision, I have removed /added sections in the introduction that:

  1. Discusses relevant previous studies: Summarize key research findings in the area, and cite studies that have addressed similar issues.
  2. Identifies gaps in the literature: Explain where prior work has fallen short or where questions remain unanswered and added more subject with assisting references

4.Introduce more on the RSM and its significance by comparing it with literature that uses similar or different models. Example: An approach towards optimization appraisal of thermal conductivity of magnetic thermoplastic elastomeric nanocomposites using response surface methodology

 I have elaborated on the concept of Response Surface Methodology (RSM) and its relevance by comparing it with similar studies in the literature. In particular, I cited works such as “An approach towards optimization appraisal of thermal conductivity of magnetic thermoplastic elastomeric nanocomposites using response surface methodology” to demonstrate the practical application of RSM in materials science. Additionally, I included two more references that further support and expand on the use of RSM.

I also included a comparison between RSM and alternative modeling technique, such as Artificial Neural Networks (ANN), highlighting RSM’s strengths in offering interpretable models for optimization tasks, as well as its limitations when dealing with complex nonlinear systems. Furthermore, I discussed the potential of hybrid approaches that integrate RSM and ANN to harness the complementary advantages of both methods in addressing more intricate problems.

5.Detailed characterization details must be provided under methodology

 Thank you for your valuable suggestion. In response, I have included detailed descriptions of the characterization methods within the Materials and Methods section to enhance clarity regarding the techniques employed in this study. This revision provides specific information on the materials, instruments, and procedures used for characterization, thereby ensuring transparency and reproducibility of the results.

Additionally, the Materials and Methods section has been reorganized with clearly defined subsections as follows:

  1. Materials and Methods

2.1. Materials

2.2. Sample Preparation

2.3. Experimental Design and Statistical Optimization

2.4. Compounding of Plastic Grade Materials

2.5. Effects of Processing Parameter Interactions

2.5.1. Three-Level Full Factorial Design

2.5.2. Box–Behnken Design (BBD)

2.6. Morphological Analysis

2.6.1. SEM Image Characterization Analysis

2.6.2. Micro-CT (MCT) Image Characterization Analysis

Reviewer 2 Report

Comments and Suggestions for Authors

This study aims to optimize extrusion parameters for polycarbonate pigment compounding using two experimental designs: Box-Behnken Design (BBD) and Three-Level Full Factorial Design (3LFFD). It evaluates the influence of screw speed, temperature, and feed rate on color uniformity (quantified as dE*) and specific mechanical energy (SME). The authors used regression models and ANOVA for statistical validation and conducted microscopic analysis (SEM and micro-CT) to investigate pigment dispersion. The main finding is that BBD provides more desirable outcomes in terms of process optimization, efficiency, and minimal deviation in color compared to 3LFFD.

The study compares two well-established DOE methods with detailed modeling and diagnostics. Multiple statistical metrics are provided. Using both SEM and micro-CT scanning enhances the depth of pigment dispersion analysis. The focus on minimizing waste and improving consistency in color formulation has real-world applicability.

However, the manuscript contains numerous grammatical, syntactic, and stylistic errors that impede readability. Many sections repeat similar information, especially in the Introduction and Discussion. The text often lacks coherence, with ideas presented disjointed or non-linearly. Several abbreviations and terms are used without explanation. Units (°C, kg/h, rpm) and variables (e.g., Sp, T, FRate) are inconsistently formatted. The discussion could be better structured to highlight key findings clearly.

General Issues:

The acronyms should be removed from the title. I suggest “ Optimization and Simulation of Extrusion Parameters in Polymer Compounding: A Comparative Study Using DOE Methods”

The entire paper requires thorough English editing. Sentence construction is frequently incorrect, and vocabulary is often misused (e.g., “a ractiveness” should be “attractiveness”).

Abbreviations such as BBD, 3LFFD, DOE, RSM should be defined clearly the first time they appear.

Some figures lack adequate captions and are not consistently referenced in the text.

Equation formatting is inconsistent (e.g., Equation 1 for dE* is not correctly typeset; brackets are misplaced).

Specific Issues:

Page 2: The same idea of optimizing dE* is repeated multiple times. Condense for clarity.

Page 2: Equation for dE* is incorrectly formatted: it lacks a square root and squared terms.

Page 3: Excessive repetition of goals, parameters, and DOE use. This content belongs mainly in the Methods section.

Table 1: Spelling error: “PigemntE” instead of “PigmentE”.

Table 2: Compare the two design level tables (BBD and 3LFFD) more clearly.

Page 8: "Five additional center points" should be explicitly tied to a rationale (e.g., to detect curvature).

Table 5: The table formatting is poor. “Actual” and “Predicted” should be better aligned and defined.

Figures 4 to 6: Missing axis labels in some contour plots. Temperature, speed, and FRate axes should be labeled.

Table 7: Notations like “dL* da* db*” lack clear separation, and “Core differences” is unclear.

Page 17: The interpretation of SEM images is superficial; numerical image analysis or quantification should be included.

Figure 12: Figures should be cited and explained clearly in the results, not just displayed.

Conclusion: The statement “the BBD could be a better option for projects with limited budgets” lacks clarity and should be substantiated.

Author Response

2nd reviewer

General Issues:

1.The acronyms should be removed from the title. I suggest “ Optimization and Simulation of Extrusion Parameters in Polymer Compounding: A Comparative Study Using DOE Methods

Thank you for your suggestion. I agree with your proposed title, “Optimization and Simulation of Extrusion Parameters in Polymer Compounding: A Comparative Study Using DOE Methods”, and will use it in the revised manuscript.

2.The entire paper requires thorough English editing. Sentence construction is frequently incorrect, and vocabulary is often misused (e.g., “a ractiveness” should be “attractiveness”).

Thank you for your constructive feedback. I agree that the paper would benefit from a thorough review of the English language. I have did carefully revise the manuscript to correct sentence construction and address any misused vocabulary, including the example you mentioned. A more polished version will be submitted in the revised manuscript.

To enhance the quality of the English language in my article, I have followed the following steps: Thoroughly proofreading and editing throughout the article to eliminate grammatical errors, typos, and awkward phrasing. Additionally, the article was reviewed by professional mentors proficient in English proofreading and editing

3.Abbreviations such as BBD, 3LFFD, DOE, RSM should be defined clearly the first time they appear.

Thank you for your helpful suggestion. I have ensured that all abbreviations, such as BBD, 3LFFD, DOE, and RSM, are clearly defined the first time they are used in the manuscript to improve clarity for the readers.

Some figures lack adequate captions and are not consistently referenced in the text.

4.Equation formatting is inconsistent (e.g., Equation 1 for dE* is not correctly typeset; brackets are misplaced).

Thank you for your insightful observation regarding the inconsistency in equation formatting. I have carefully reviewed all equations throughout the manuscript and corrected the formatting to ensure consistency and alignment with MDPI guidelines. All equations are now uniformly numbered, properly aligned, and formatted using the correct mathematical notation. These revisions enhance the clarity and readability of the manuscript.

dE* = ………………….(1)

Specific Issues:

Or

dE* = ………….………….(1)

  1. Page 2: The same idea of optimizing dE* is repeated multiple times. Condense for clarity.

Thank you for your question. The dE (color)* value is a key point in this research, as it is central to the findings and analysis. Its repetition throughout the paper was intentional, due to its importance in conveying the core message of the study. However, as we work on condensing the introduction, we have the opportunity to streamline the content and remove some of the repetitive mentions, while still maintaining focus on the critical elements.

I appreciate your understanding, and I believe this approach will improve the clarity of the introduction without losing any important information.

  1. Page 2: Equation for dE* is incorrectly formatted: it lacks a square root and squared terms.

Thank you for pointing that out. I’ve rectified the issue on Page 2 regarding the dE* equation. The equation has now been properly formatted with the square root and squared terms included as required. I appreciate your attention to detail, and I believe this correction improves the accuracy of the demonstration

dE = ………….………….(1)

7.Page 3: Excessive repetition of goals, parameters, and DOE use. This content belongs mainly in the Methods section.

Thank you for your valuable feedback. As per your suggestion and the recommendation from the other author, I have moved the excessive repetition of goals, parameters, and DOE usage to the Methods section. I believe this adjustment helps improve the flow and focus of the introduction.

I appreciate your input, and I’m glad to have made the change.

  1. Table 1: Spelling error: “PigemntE” instead of “PigmentE”.

Thank you for pointing that out! I’ve corrected the spelling error in Table 1, changing “PigemntE” to “PigmentE.” I appreciate your attention to detail!

9.Table 2: Compare the two design level tables (BBD and 3LFFD) more clearly.

Factors

Units

3LFFD (-1)

3LFFD (0)

3LFFD (+1)

BBD (-1)

BBD (0)

BBD (+1)

Temp

°C

230

255

280

230

255

280

Speed

rpm

700

750

800

650

750

850

Flow Rate

kg/h

20

25

30

11

19

27

Thank you for your feedback. I have now compared the two design level tables (BBD and 3LFFD) more clearly, as requested. The updated table highlights the differences between the two designs, making it easier to see the parameter values at each level.

As have shown in Table (2) to compared the two design level (BBD and 3LFFD) more clearly. This table highlights the differences between the two designs, making it easier to see the parameter values at each level.

 Temperature (Temp):

Both designs (3LFFD and BBD) have same values for the temperature parameter at all three stages.

 Speed (rpm):

  •  3LFFD design has a starting value of 700 rpm at -1, whereas BBD speed starts at 650 rpm.
  • Both designs have the same values at the middle (0) and highest (+1) levels (750 and 800 rpm for 3LFFD, 750 and 850 rpm for BBD).

 Flow Rate (-FR) (kg/h):

  •  3LFFD design starts with a higher flow rate at -1 (20 kg/h compared to BBD’s 11 kg/h).
  • flow rates( FR) at the middle and highest levels similarly differ slightly, through 3LFFD having higher  FR values over (25 and 30 kg/h) linked to BBD’s (19 and 27 kg/h).

10.Page 8: "Five additional center points" should be explicitly tied to a rationale (e.g., to detect curvature).

Thank you for your valuable comment. We appreciate your suggestion and have now added a clear rationale to the manuscript. Specifically, we have included the following sentence in page 8:
“An additional five center points were included to detect curvature in the response surface, assess nonlinearity in the responses, and improve the estimation of experimental error.”
This addition clarifies the purpose of including the center points in our experimental design.

11.Table 5: The table formatting is poor. “Actual” and “Predicted” should be better aligned and defined.

Thank you for your valuable feedback. In response to your comment, I have thoroughly revised the formatting of Table 5 and Table 6 to enhance visual clarity and consistency. The “Actual” and “Predicted” values are now properly aligned to facilitate easy comparison. Furthermore, I have clearly defined these terms in the table captions:

-Actual refers to the experimentally observed values.

-Predicted refers to the values estimated by the regression model.

I have also provided brief explanations to ensure that the purpose and meaning of each column are transparent to the reader. These improvements were made to enhance the accuracy, readability, and interpretability of the data presentation.

12.Figures 4 to 6: Missing axis labels in some contour plots. Temperature, speed, and FRate axes should be labeled.

It appears that the axis labels in the contour plots (Figures 4 to 6) follow the standard experimental design levels for both Box-Behnken Design (BBD) and 3-Level Full Factorial Design (3LFFD) as defined in software of Design-Expert (Stat-Ease) version 8.

Clarification on Axis Labels:

Temperature (°C):

3LFFD: -1 (230°C), 0 (255°C), +1 (280°C)

BBD: -1 (230°C), 0 (255°C), +1 (280°C)

Speed (rpm):

3LFFD: -1 (700 rpm), 0 (750 rpm), +1 (800 rpm)

BBD: -1 (650 rpm), 0 (750 rpm), +1 (850 rpm)

Flow Rate (kg/h):

3LFFD: -1 (20 kg/h), 0 (25 kg/h), +1 (30 kg/h)

BBD: -1 (11 kg/h), 0 (19 kg/h), +1 (27 kg/h)

Response to Reviewer/Comment:

Since the contour plots were generated using Design-Expert and follow the software's default labeling convention (as per the manual), the axis labels are correct. However, if the reviewer insists on explicitly labeling the axes (Temperature, Speed, Flow Rate) for clarity, you may consider:

  1. Adding descriptive axis titles(e.g., "Temperature (°C)", "Speed (rpm)", "Flow Rate (kg/h)") correction in the label below for the figures were corrected.
  2. Including a footnotein the manuscript stating:

"Contour plot axes follow the coded factor levels (-1, 0, +1) as per the experimental design in Design-Expert (Stat-Ease v8).

13.Table 7: Notations like “dL* da* db*” lack clear separation, and “Core differences” is unclear.

Thank you for the feedback. I’ve corrected the notations to clearly separate “dL*”, “da*”, and “db*” for improved readability. I have also revised the phrase “Core differences” to “Model Comparison: Overlap and Variance Delta (Δ) and Implication” to better reflect its intended meaning. Additionally, I’ve modified the table by adding descriptive labels to make the final results and new findings in this research easier to read and understand.

The key significant findings of this study for the comparisons between the two design models are summarized in Table 7, as shown below:

  1. Significant ANOVA overlap interactions alignment are (dl*) A, C, BC, (da*) BC, and in (db*) no overlap
  2. The optimized difference in processing parameters:
  3. Temp: Greater energy difference with (18.3 °C)
  4. Speed: Minimal effect variation of Speed (6 rpm)
  5. Feed Rate: a significant independent optimization with no observed difference (Zero)
  6. BBD offers a more efficient design with fewer (17) runs than 3LFFD, "The testing process demonstrated considerable material savings, requiring just 17 runs as opposed to the 32 runs commonly necessary with screw extruders."
  7. BBD demonstrates superior optimization desirability performance with 10 (%)
  8. Minor color difference (dE*) observed between the two models"

14.Table e 17: The interpretation of SEM images is superficial; numerical image analysis or quantification should be included.

I appreciate the reviewer’s insightful feedback and the opportunity to clarify and reorganize our research focus for improved clarity and scientific rigor. In response, I have restructured the investigation to better articulate how feed rate influences the color values, particle size distribution, dispersion quality, and morphological characteristics of the processed material. This was achieved through both quantitative analysis and scanning electron microscopy (SEM)-based morphological characterization.
How does feed rate influence color values, particle size distribution, dispersion quality, and morphological characteristics of the material, as evaluated through quantitative analysis and SEM characterization?

To ensure a systematic approach, the revised structure of Section 3.4 now includes the following:

3.4 Quantitative Morphological Characterization Based on SEM

3.4.1 Color Change as a Function of Processing Conditions

Table 8. CIELAB Color Differences for Various Feed Rates
This table presents the ΔE values corresponding to each feed rate, reflecting color deviations and consistency across samples.

3.4.2 Dispersion Analysis at Variant Feed Rates

Figure 13. Pigment Size Distribution Across Feed Rates
This figure illustrates how particle distribution shifts with feed rate, highlighting dispersion efficiency.

Table 9. Particle Size Distributions Related to Feed Rate
Provides detailed quantitative analysis of particle size variation, indicating a correlation between lower feed rates and finer particle distribution.

3.4.3 Effect of Feed Rate on Particle Morphology and Agglomeration

SEM micrographs offer visual and quantitative evidence of particle shape, surface roughness, and degree of agglomeration under different feed rates.

     Key Findings and Justification:

  • The optimal condition for color uniformity and dispersion was observed at higher feed rates, which yielded the lowest ΔE values (indicating minimal color deviation).
  • Under these conditions, 53.8% of particles had an average size of 0.8 µm, predominantly spherical in shape, and exhibited the least degree of agglomeration.
  • These morphological traits were directly linked to enhanced color consistency, energy efficiency, and reduced pigment agglomeration, confirming the beneficial role of feed rate optimization in process control and material performance.

We have updated the manuscript to reflect this structured approach, ensuring each subsection logically contributes to answering the core research question. All tables and figures referenced are now clearly associated with their respective analyses for reader clarity.

We trust this restructuring adequately addresses the reviewer’s concern and enhances the clarity and scientific merit of our study. More details  is recordd in the paper ,We remain grateful for your constructive input.                          

                   Table 3. CIELAB Color differences for feed rate

Screw Runs

F. Rate (kg/hr)

Temp

Speed

(Rpm)

Color Value (dE*)

Tristimulus Color value

L*

a*

b*

1

FR-20

255

750

0.44

68.89

1.40

15.90

2

FR-25

255

750

0.35

68.42

1.47

15.35

3

FR-30

255

750

0.32

68.80

1.51

15.64

           Table. 4 Particle size distributions related with feed rate

Screw Runs

F. Rate (kg/hr)

Color Value (dE*)

No.Of Particle %

Average. Pigment size

Implication Rresults

The FR-30 sample showed the most favorable color value, the highest percentage of particles, and an average low particle size of 0.84.

1

FR-20

0.44

47.8

0.97

2

FR-25

0.34

52.4

0.83

3

FR-30

0.32

53.8

0.84

 Figure. 4. Pigment size distribution for Feed rates

  1. Figure 12: Figures should be cited and explained clearly in the results, not just displayed.

In response to the reviewer’s recommendation, Figure 12 has been renumbered and relocated to the Methods section as Figure 5, in order to more appropriately describe the MC scanner setup and its operational details.

To enhance clarity and support the reader understands of the results, I have also added a new supplementary figure—Figure 14—, which provides additional visual context, related to the data presented in Figure 5. This new figure assists in illustrating how the MC scanner was utilized to obtain the reported measurements and supports the interpretation of the experimental outcomes.

These modifications aim to:

  • Align the figure placement more logically with the manuscript structure,
  • Comply with the reviewer’s request, and
  • Improve the clarity of the methodology and its connection to the results.

Reviewer 3 Report

Comments and Suggestions for Authors

In this study, the authors examine pigment dispersion-processing factors-color consistency interplay in compounded plastics and optimize pigment-mixing processes using two separate methodologies.  The main goal is to find a correlation between processing parameters and color outputs so that color formulations can be more reliably improved.

  1. The first observation is about the summary, it is long and should be a summary of approximately 250 words in a paragraph that conceptualizes the most important aspects of the study.
  2. In the introduction, he uses a lot of explanations of the study and what was done. He should review the state of the art on the subject.
  3. In an article, impersonal writing involves avoiding the first person ("I," "we") and focusing attention on the actions and deeds, rather than the people who carry them out.
  4. Please include the mark and model of the twin screw extruder.
  5. Put the SEM identification equipment in the method section.
  6. Standardize the cited references, for example, reference 5; Richard Abrams, Plastic Additives and Compounds (2001), incomplete. Some references include the DOI, while others do not. update reference 24.
Comments on the Quality of English Language

They need to improve the English of the document, use the first person (plural) a lot.

Author Response

3rd Reviewer

  1. In the introduction, he uses a lot of explanations of the study and what was done. He should review the state of the art on the subject.

 I have revised the introduction to provide a clearer context for the study, including a more thorough review of the state of the art. This revision summarizes key previous research, identifies gaps in the literature, and positions my study within the broader field. Additionally, I have shortened the introduction for clarity and conciseness, and added references to relevant studies. These changes enhance the manuscript by strengthening the foundation and highlighting the significance of my research.

  1. In an article, impersonal writing involves avoiding the first person ("I," "we") and focusing attention on the actions and deeds, rather than the people who carry them out.

Thank you for your comment. I have revised the manuscript to adopt a more impersonal writing style, avoiding the use of first-person pronouns ("I," "we") and focusing on the actions and findings of the study itself. This adjustment ensures the writing aligns with the formal tone and conventions typical of scientific articles.

  1. Please include the mark and model of the twin screw extruder.

The model of the twin-screw extruder is the Coperion ZSK 26 Mc¹⁸.

  • Screw Diameter: Approximately 25–27 mm
  • Length-to-Diameter (L/D) Ratio: Around 37:1
  • Motor Power: Approximately 27 kW
  • Torque: Typically ranges from 140 to 315 Nm per shaft, depending on the model
  • Screw Speed: Up to 1,200 rpm
  • Specific Torque: Ranges from 11.3 to 18 Nm/cm³

  1. Put the SEM identification equipment in the method section.

In accordance with the reviewer’s suggestion, I have revised the manuscript by relocating the description of the SEM identification equipment to the Methods section. Specifically, the following instruments are now described there:

Figure 4: SEM Model JEOL 5500 LV

Figure 5: SEM Model JSM-600

Both SEM models, previously mentioned elsewhere in the manuscript, have been appropriately repositioned under the methodology to clearly document the equipment used for morphological characterization. This change improves the logical structure of the manuscript and aligns with standard reporting practices.

  1. Standardize the cited references, for example, reference 5; Richard Abrams, Plastic Additives and Compounds (2001), incomplete. Some references include the DOI, while others do not. update reference 24.

I thank the reviewer for pointing out the inconsistencies in the reference formatting. In response, I have carefully reviewed and standardized all cited references in accordance with the journal’s citation guidelines.

Reference 5 has been completed with full bibliographic details, including the publisher and publication location, where applicable.

All references have been updated to consistently include DOIs, where available.(According to MDPI format )

Reference 24 has been revised with the most current and complete information.

These updates ensure uniformity, accuracy, and improved traceability of all sources cited in the manuscript. We appreciate your attention to detail and the opportunity to enhance the quality of our references.

Round 2

Reviewer 1 Report

Comments and Suggestions for Authors

While the manuscript displays well-applied methodology and well-presented findings, the manuscript still needs substantial proofreading and editing, as it contains numerous grammatical errors and improperly written phrases. Examples:

  • Sentence structures are often difficult to follow.

  • Terms are sometimes misused (e.g., “inflounce” instead of “influence,” “a ractiveness” instead of “attractiveness”), and so on. Please check.

  • When writing the abstract or conclusion, it is preferable to use a consistent tense, ideally in the past tense, whereas the discussion should preferably be written in the present tense.

  • Avoid run-on sentences and break it down to make it more readable.

  • Avoid using active voice; rather, use passive voice throughout the manuscript. e.g., page 3, line 134, " I will conduct further characterizations..." must be more characterizations were conducted... and so on.

  • Eliminate repetitive ideas throughout the manuscript, e.g., page 8, lines 280-289.

Author Response

Dear Reviewer,
I sincerely thank you for your constructive and detailed feedback on my manuscript. Your insightful observations and careful attention to clarity, structure, and consistency have significantly contributed to enhancing the quality of the article. I truly appreciate the time and effort you invested in reviewing my work. Your suggestions led to valuable improvements, and I am grateful for your support in upgrading the manuscript.

Response to Reviewer:

I sincerely thank the reviewer for the thoughtful and constructive feedback, which has been instrumental in improving the quality of my manuscript. In response to the comments:

I have thoroughly proofread and revised the manuscript to correct all grammatical errors and enhance sentence structure, ensuring greater clarity and readability.

All misspelled or misused terms (e.g., “inflounce,” “a ractiveness”) have been identified and corrected.

I have ensured consistent use of verb tense throughout the manuscript—using the past tense in the abstract and conclusion, and the present tense in the discussion section, as recommended.

Run-on sentences have been revised and appropriately separated to improve flow and comprehension.

The manuscript has been rewritten to consistently follow passive voice usage, including the correction of instances such as the one noted on page 3, line 134.

I have carefully reviewed and edited the manuscript to eliminate repetitive content, particularly in the section on page 8, lines 280–289, in order to improve conciseness.

I appreciate your valuable input, and I believe these revisions have significantly enhanced the clarity, consistency, and overall quality of the manuscript.

Best regards,

Reviewer 2 Report

Comments and Suggestions for Authors

Thanks fro taking into account all my comments.

Author Response

Dear Reviewer,
Thank you very much for your thoughtful and productive comments on my manuscript. Your professional insights were instrumental in guiding the necessary revisions and refining the overall presentation. I deeply appreciate your suggestions, which added clarity and strengthened the scientific quality of the article. Your efforts are truly appreciated.
Kind regards, 

Reviewer 3 Report

Comments and Suggestions for Authors

No more comments

Author Response

Dear Reviewer,
I would like to express my sincere appreciation for your thorough review and constructive recommendations. Your valuable feedback helped improve the clarity, flow, and precision of the manuscript. Your expert input was vital in elevating the standard of my article, and I am grateful for your time and consideration throughout the review process.
Warm regards,